# Therapeutic Applications for Oncolytic Self-Replicating RNA Viruses

**DOI:** 10.3390/ijms232415622

**Published:** 2022-12-09

**Authors:** Kenneth Lundstrom

**Affiliations:** PanTherapeutics, Route de Lavaux 49, CH1095 Lutry, Switzerland; lundstromkenneth@gmail.com

**Keywords:** recombinant viral particles, RNA replicons, DNA replicons, oncolytic viruses, cancer vaccines, cancer immunotherapy, preclinical studies, clinical trials

## Abstract

Self-replicating RNA viruses have become attractive delivery vehicles for therapeutic applications. They are easy to handle, can be rapidly produced in large quantities, and can be delivered as recombinant viral particles, naked or nanoparticle-encapsulated RNA, or plasmid DNA-based vectors. The self-replication of RNA in infected host cells provides the means for generating much higher transgene expression levels and the possibility to apply substantially reduced amounts of RNA to achieve similar expression levels or immune responses compared to conventional synthetic mRNA. Alphaviruses and flaviviruses, possessing a single-stranded RNA genome of positive polarity, as well as measles viruses and rhabdoviruses with a negative-stranded RNA genome, have frequently been utilized for therapeutic applications. Both naturally and engineered oncolytic self-replicating RNA viruses providing specific replication in tumor cells have been evaluated for cancer therapy. Therapeutic efficacy has been demonstrated in animal models. Furthermore, the safe application of oncolytic viruses has been confirmed in clinical trials. Multiple myeloma patients treated with an oncolytic measles virus (MV-NIS) resulted in increased T-cell responses against the measles virus and several tumor-associated antigen responses and complete remission in one patient. Furthermore, MV-CEA administration to patients with ovarian cancer resulted in a stable disease and more than doubled the median overall survival.

## 1. Introduction

Cancer still remains the leading cause of worldwide mortality, with 10 million deaths annually [1]. Despite progress in diagnostics and therapy, the incidence and mortality numbers remain high due to pollution, unhealthy eating habits, lifestyle choices, and an aging population [2]. Although progress in conventional chemotherapy and radiotherapy approaches have been made, the efficient and safe delivery of cancer drugs has been a major obstacle. In this context, both nonviral and viral delivery vectors have been engineered for cancer therapy in parallel to conventional approaches [3].

Different applications of viral vectors have been used for the development of cancer vaccines and therapy, focusing on the overexpression of tumor-associated antigens (TAAs), anticancer genes, and immunostimulatory genes [4,5]. One relatively novel approach comprises the use of oncolytic viruses, which specifically kill tumor cells without causing damage to normal tissue due their targeted replication in tumor cells [6]. Oncolytic viruses exist as naturally occurring [7] and engineered [8] versions. Different types of viruses such as adenoviruses [9], alphaviruses [8], Herpes simplex viruses [10], rhabdoviruses [11], Newcastle disease virus [12], and vaccinia viruses [13] have demonstrated oncolytic properties. However, in this review, the focus is entirely on oncolytic viruses belonging to the families of self-replicating RNA viruses such as alphaviruses [8], flaviviruses [14], measles viruses [15], and rhabdoviruses [11]. The self-replication of viral RNA and the application of oncolytic self-replicating RNA viruses in preclinical animal models and in clinical trials on cancer patients are described.

## 2. Characterization of Oncolytic Self-Replicating RNA Viruses

Studies on the origin of cancer have indicated that a subpopulation of cells known as cancer stem cells (CSCs) or cancer-initiating cells (CICs) are responsible for tumorigenesis [16]. As CICs have been shown to be resistant to conventional anticancer therapies, the potential of oncolytic viruses to destroy CICs have made them attractive for alternative therapeutic applications. Oncolytic viruses of different origin [7,8,9,10,11,12,13,14,15] comprise wild-type viruses, which are unable to infect normal cells but are cytotoxic to cancer cells [17]. Moreover, the deletion of viral genes critical for replication in normal cells but dispensable in cancer cells has generated attenuated oncolytic strains. Serial passaging in cell cultures has also resulted in attenuated viruses. The mechanisms have been postulated to involve RAS pathway activation or take place by genetic modifications [18]. For these reasons, oncolytic viruses present efficient tumor killing, while only minimal toxicity is caused in normal cells.

Self-replicating RNA viruses possess a special feature in the ability of self-replicating of their RNA genome in infected host cells, resulting in approximately 200,000-fold RNA amplification [19]. The single-stranded RNA (ssRNA) genome is of positive polarity for alphaviruses [19] and flaviviruses [20]. In contrast, measles viruses [21] and rhabdoviruses [22] possess a negative-stranded genome. This difference is significant, as, in the former case, viral RNA can directly be translated in the cytoplasm of infected cells, whereas, in the latter case, positive-stranded copies need to be generated prior to translation.

Among alphaviruses, the naturally oncolytic M1 alphavirus has been used in several cancer therapeutic applications [23,24]. Moreover, attenuated Sindbis virus (SIN) strains such as SIN AR339 [25] and vectors based on the Semliki Forest virus (SFV) strain SFV-A7(74) [26] have demonstrated oncolytic properties. Additionally, Aura virus (AURAV) has shown oncotropism for certain tumor cell lines [27]. In the context of flaviviruses, the Zika virus (ZIKV) has demonstrated oncolytic activity against glioblastoma stem cells (GSCs) [14,28]. The negative-stranded measles viruses (MV) have also demonstrated oncolytic activity in several preclinical studies [15]. In the case of rhabdoviruses, the vesicular stomatitis virus (VSV) has been utilized for cancer therapy due to its oncolytic activity [11,29]. Moreover, the oncolytic Maraba virus has been used for the treatment of sarcoma [30]. The delivery of self-replicating RNA viruses is illustrated in Figure 1. 

## 3. Preclinical Studies Using Oncolytic Self-Replicating RNA Viruses

Due to the large number of preclinical studies conducted with oncolytic self-replicating RNA viruses, selected examples for studies using alphaviruses, flaviviruses, measles viruses, and rhabdoviruses are presented below and summarized in Table 1.

In the context of alphaviruses, the oncolytic replication-proficient SFV (A774nsP) strain was engineered to express enhanced green fluorescent protein (VA7-EGFP) and Renilla luciferase (VA7-Rluc) [31]. VA7-EGFP efficiently infected and killed human U87, U251, and A172 glioma cells. Intravenous administration of VA7-EGFP completely eradicated 100% of small and 50% of large subcutaneous U87 tumors in BALB/c mice [31]. Moreover, long-term survival was established in 16 out of 17 mice after a single VA7-EGFP or VA7-Rluc injection. In another study, the VA7-EGFP vector was evaluated in an orthotopic A549 lung cancer model in nude mice [26]. Local administration of VA7-EGFP showed strong responses, and no large tumor masses were detected. Moreover, the survival rate of mice increased significantly and was similar to that seen for treatment with the conditionally replicating Ad5-Delta24TK-GFP adenovirus vector. The VA7-EGFP vector has also been tested in human VCaP, LNCaP, and 22Rv1 prostate tumor cell lines, showing efficient cell killing [32]. In contrast, the benign RWPE-1 prostate epithelial cell line was highly resistant to VA7-EGFP. Moreover, sufficient eradication of subcutaneous and orthotopic LNCaP tumors was observed after a single peritoneal injection. The oncolytic activity of SFV hampered by antiviral signaling by type I interferon (IFN-I) has been counteracted by engineering the IFN-I-resistant SFV-AM6 vector [33]. Despite high IFN-1 signaling in mouse GL261 cells, SFV-AM6 induced immunogenic apoptosis. Furthermore, the introduction of microRNA-124 (miR124) into the SFV-AM6 vector provided selective replication in glioma cells and infection of orthotopic GL261 gliomas after intraperitoneal administration in mice [33]. Moreover, combining of SFV-AM6-124T with anti-programmed death 1 (PD1) immunotherapy enhanced the infiltration of immune cells in GL261 gliomas. In another study, the SFV4miRT vector containing miR124, miR125, and miR134 was engineered to reduce the neurovirulence of the vector [34]. SFV4miRT provided replication in mouse neuroblastoma and glioblastoma cell lines. Moreover, a single intravenous administration of SFV4miRT resulted in prolonged survival and the cure in 4 out of 8 mice with NX52 xenografts and 3 out of 11 with CT-2A xenografts [34]. In contrast, no cure was observed in mice carrying GL261 tumors. The oncolytic SIN AR339 strain showed cytopathic effects and apoptosis in cervical HeLaS3 and C33A cell lines and ovarian HOC-1, HAC-2, and OMC-3 cancer cell lines [25]. However, similar effects were not observed in human keratinocytes. The intratumoral and intravenous administration of SIN AR339 generated significant regression of cervical tumors in mice [25]. Furthermore, the suppression of ascites formation was detected in a metastasis model of ovarian cancer after the intraperitoneal injection of SIN AR339.

Among naturally oncolytic alphaviruses, M1 has demonstrated selective infection and killing of zinc finger antiviral protein (ZAP)-deficient tumor cells, causing hardly any damage to normal cells [7]. Introduction of the GFP gene into the M1 vector allowed in vitro and in vivo visualization, demonstrating stable GFP expression for at least 10 generations in colorectal cancer cells and tumor targeting in BALB/c mice with Hep3B liver tumor xenografts [35]. Investigation of the biodistribution in mice, rats, and cynomolgus macaques carrying tumors demonstrated a gradual elimination of M1 in normal tissue, while prominent M1 replication increased in tumor tissue [36]. Additionally, the efficient killing of malignant glioma cells was detected due to M1 infiltration of the blood–brain barrier. In another approach, cell viability was significantly decreased in eight different bladder cancer cell lines after M1 infection, whereas no effect was seen in normal bladder cells [37]. Replication of M1-GFP occurred in T24 and UM-UC-3 bladder tumor cell lines and in primary tumor cells from patients but not in normal bladder cells [37]. Moreover, a significant inhibition of tumor growth was seen in mice implanted with muscle-invasive bladder cancer (MIBC) after tail vein injections of M1-GFP. Furthermore, M1 showed oncolytic activity in an orthotopic mouse bladder model [38]. Suppression of the coiled-coiled domain containing 6 (CCDC6) gene in M1-infected mice provided enhanced oncolytic activity of M1. In the case of breast cancer, M1 showed oncolytic activity against highly aggressive triple-negative breast cancer (TNBC) in vitro, which was enhanced 100-fold by doxorubicin combination therapy and slowed down tumor growth in vivo [39]. In another combination therapy approach, irreversible electroporation (IRE) and M1 showed a synergistic effect in pancreatic cancer cell lines [40]. The M1-IRE combination resulted in superior inhibition of tumor proliferation and prolonged survival in immunocompetent mice carrying orthotopic pancreatic tumors. In another approach, M1 particles were encapsulated in liposomes (M-LPO), which inhibited the growth of colorectal LoVo and liver Hep3B cancer cells [41]. In contrast, naked M1 particles, liposomes, or a mixture of M1 particles and liposomes did not inhibit tumor growth. It was also demonstrated that intravenous M-LPO injections into mice reduced the production of M1-specific neutralizing antibodies, which may reduce the intrinsic viral immunogenicity, providing superior oncolytic activity.

Among flaviviruses, ZIKV, known for causing cell death and the differentiation of neural precursor cells in a developing fetus, has also been characterized for its oncolytic activity against glioblastomas [28]. ZIKV has shown preferential infection and killing of GSCs, which has been confirmed by the depletion of patient-derived GSCs grown in culture and organoids. Administration of a mouse-adapted ZIKV strain to mice with implanted glioblastomas resulted in substantially prolonged survival [28]. It has also been confirmed that ZIKV preferentially infects and kills GSCs and stem-like cells in medulloblastoma and ependymoma in a SRY (sex-determining region Y)-box2 (SOX2)-dependent manner [42]. Moreover, a negative correlation with the expression of antiviral interferon response genes (ISGs) and a positive association with integrin α_v_ were detected. ZIKV showed selective elimination of GSCs from human mature cerebral organoids and GBM surgical specimens, suggesting the potential of using ZIKV for brain tumor therapy. In another study in C57BL6/J mice, coadministration of ZIKV improved the moderate tumor survival of anti-PD1 antibody monotherapy [43]. Furthermore, ZIKV-mediated tumor clearance provided a durable protection against syngeneic tumor rechallenges. The Brazilian ZIKV^BR^ strain has also been evaluated for its oncolytic activity against embryonal CNS tumors [44]. The investigation of different administration routes and the number of injections of ZIKV^BR^ on tumor tropism, tumor eradication, and side effects demonstrated a tropism to the brain after systemic delivery in mice with subcutaneous tumors. In the case of brain tumors, systemic administration of ZIKV^BR^ resulted in efficient tumor eradication and increased survival without any neurological or other organ injury [44]. In another study, the safety and therapeutic activity of the intrathecal administration of ZIKVBR were evaluated in three dogs with spontaneous advanced stage brain tumors [45]. No negative clinical adverse events were registered. Moreover, reduced tumor size, significantly improved neurological clinical symptoms, and prolonged survival were observed. ZIKV^BR^ also induced eradication of tumor cells and activation of immune responses without causing damage to normal cells. The oncolytic potential of ZIKV has also been evaluated for prostate cancer [46]. In a metabolomic approach, 21 statistically relevant markers from prostate PC-3 cancer cells infected by the inactivated ZIKV prototype (ZVp) were identified [47]. It was discovered that the markers were associated with lipid remodeling, endoplasmic reticulum stress, inflammatory mediators, and disrupted porphyrin and folate metabolism. 

Clinical observations from patients with leukemia and lymphoma from the 1970s and 1980s demonstrated that MV infections can induce oncolytic immunotherapeutic responses [15]. More recently, expression of GFP from an oncolytic MV strain resulted in complete regression of tumors after a single intratumoral administration in a medulloblastoma mouse model [47]. In another study, intracerebral injection of MV-GFP into mice with medulloblastoma prolonged their survival significantly [48]. Moreover, oncolytic MV strains expressing the carcinoembryonic antigen (CEA) and sodium iodide symporter (NIS) have been engineered, demonstrating infection, replication, and cytopathic effects in GSC cell lines [49]. Moreover, when MV-GFP-infected GBM44 GSCs were implanted into the right caudate nucleus of nude mice, a significant prolongation of survival was seen [49]. Related to breast cancer, the engineered rMV-SLAMblind vector carrying the signaling lymphocyte activation molecule (SLAM) targeting the poliovirus receptor-related 4 (PVRL4), showed antitumor activity against mice implanted with human breast cancer xenografts [50]. Moreover, a live attenuated MV effectively infected and killed human MCF-7 and CAL-51 breast cancer cell lines [51]. In another study, the MV Edmonston strain (MV-Edm) provided re-sensitization of breast cancer cells to doxorubicin and ionizing radiation, potentially resulting in superior therapeutic efficacy [52]. In the context of lung cancer, the oncolytic MV Hu-191 strain suppressed tumor growth and prolonged survival in C57BL/6 mice [53], and the oncolytic MV Schwarz strain repressed growth of lung and colorectal tumors in nude mice [54]. Moreover, the MV-Edm-CEA vector killed non-small-cell lung cancer (NSCLC) cell lines in culture and provided tumor regression in intratumorally injected nude mice with A549 tumors [55]. Related to melanoma, the MV Leningrad-16 (L-16) strain infected and killed melanoma cell lines efficiently and inhibited tumor growth in the melZ mouse melanoma model [56]. Pancreatic cancer has also been targeted by MV vectors. For example, intratumoral injection of the MV-SLAMBlind vector resulted in significant inhibition of tumor growth in mice implanted with KLM1 and Capan-2 pancreatic tumor xenografts [57]. Moreover, combination therapy of the oncolytic MeV-SCD vector and gemcitabine in human M1A, Paca-2, PANC-1, and BxPC-3 pancreatic cells lines resulted in a more than 50% reduced tumor mass and only a minor effect of gemcitabine on viral replication [58]. In another approach, enhanced tumor-specific targeting of oncolytic MV vectors was achieved by introduction of the synthetic miR-148 for the regulation of vector tropism [59]. Furthermore, the targeted MV vector was combined with 5-fluorocytosine treatment, which resulted in delayed tumor growth and prolonged survival in mice with pancreatic tumor xenografts. In the context of prostate cancer, a substantial delay of tumor growth and prolonged survival were achieved in mice implanted with PC-3 prostate tumors after intratumoral injection of MV-CEA vectors [60]. Furthermore, the therapeutic efficacy of an MV vector expressing a single-chain antibody (sc-Fv) specific for the extracellular domain of the prostate-specific membrane antigen (PSMA) was evaluated in mice carrying LNCaP and PC3-PSMA prostate tumors [61]. The treatment resulted in specific infection and killing of PSMA-positive prostate cancer cells, which was enhanced by radiation therapy. Finally, combination therapy with oncolytic MV and mumps virus (MuV) vectors in PC-3 prostate tumor-bearing mice generated superior antitumor activity and prolonged survival in comparison to treatment with either MV or MuV vectors alone [62]. 

Among rhabdoviruses, the majority of oncolytic vectors are based on VSV [11,29], although the Maraba virus [30] has been used to some extent. For example, the replication competent VSVrp30a vector showed rapid tumor targeting and eradication of glioblastomas and mammary tumors in mouse models [63]. Moreover, intranasal administration of VSVrp30a showed selective infection and killing of olfactory bulb tumors without causing damage to normal cells. In another approach, systemic administration of the attenuated VSV-p1-GFP vector killed human U87 glioblastoma cells without causing any adverse neurological effects in mice [64]. The oncolytic VSV(M51R)-LacZ vector with a mutation in the matrix (M) protein was evaluated for intravenous administration in BALB/c mice carrying 4T1 mammary tumors, which resulted in infection of multiple breast cancer lesions but no cytotoxicity in normal cells [65]. In another study, intraperitoneal administration of VSV(M51R) into BALB/c mice with luciferase-expressing CT-26 tumors reduced luciferase expression and prolonged survival [66]. Moreover, administration of the chimeric VSV-GP vector expressing the glycoprotein (GP) from the lymphocytic choriomeningitis virus (LCMV) showed replication in tumor cells, tumor-to-tumor spread of the virus, and widespread killing of tumor cells in nude mice with subcutaneous LLC1 lung tumors [67]. The VSV-LCMV-GP vector has also been evaluated in subcutaneous A375 and B16-OVA syngeneic melanoma tumor models, which showed significant tumor regression and prolonged survival in mice [68]. The VSV-LCMV GP vector has demonstrated oncolytic activity in several ovarian cancer cell lines [69]. Additionally, tumor regression was seen in subcutaneous and orthotopic A2780 ovarian cancer mouse models after administration of VSV-LCMV GP. The therapeutic effect was further enhanced when VSV-LCMV GP was combined with the JAK1/2 inhibitor ruxolitinib [69]. Furthermore, subcutaneous administration of an oncolytic singe-replication cycle of the VSV vector in C57BL/6 mice implanted with B16ova melanomas resulted in strong tumor regression [70]. However, the rapid clearance of the virus suggested that the therapeutic efficacy was not dependent on VSV undergoing progressive rounds of replication. In another approach, a DNA plasmid, including the VSV M protein encapsulated in liposomes, was transfected into SKOV3 ovarian cancer cells, resulting in apoptosis [71]. Moreover, intraperitoneal administration of VSVMP-p reduced the tumor weight by 90% and prolonged the survival in carcinomatosis ovarian mouse models. Moreover, 87–98% inhibition of tumor growth and prolonged survival were detected in nude mice implanted with A2780s and A2780cp ovarian tumors after administration of VSVMP-p [72]. In another approach, it was demonstrated that pretreatment with curcumin prior to VSV(M51R) administration induced oncolysis in the PC-3 prostate cancer cell line and in a prostate cancer mouse model [73]. In the context of the Maraba virus, the oncolytic MG1 strain was engineered to express human dopachrome tautomerase (hDCT) [74]. However, MG1-hDCT neither induced antitumor immunity nor showed therapeutic efficacy in mice with B16-F10 metastases. In contrast, Maraba MG1 elicited DCT-specific responses in mice immunized with an adenovirus expressing hDCT [30]. In another study, the Maraba MG1 strain was evaluated in human and canine sarcoma cell lines and ex vivo in human sarcoma specimens and compared to reovirus, vaccinia virus, and herpes simplex virus delivery [75]. In cell lines, Maraba MG1 showed the highest potency and ex vivo infected more than 80% of human sarcoma tissues. Intratumoral administration of Maraba MG1 into BALB/c mice implanted with S180 sarcomas elicited memory-immune responses, protected mice against subsequent tumor challenges, and provided long-lasting cures in sarcoma-bearing mice. 

An interesting approach for oncolytic virotherapy comprise the engineering of photo-controllable MV and Rabies virus (RABV) vectors [75]. The positive Magnet (pMag) and negative Magnet (nMag), which heterodimerize after blue light irradiation, were introduced into the flexible domains of the L polymerase of MV and RABV. The systems were evaluated in BALB/c nu/nu mice implanted with MDM-MB-468 breast tumors. Treatment with rMV^EGFP^-L_DMH_ showed a substantial reduction in tumor growth and activation only under blue light when the oncolytic activity was switched on. 

## 4. Clinical Trials Using Oncolytic Self-Replicating RNA Viruses

The number of clinical trials conducted for oncolytic self-replicating RNA viruses is certainly much lower than those executed in animal models (Table 2). CEA has presented a popular target for clinical trials, including the utilization of both MV and the alphavirus Venezuelan equine encephalitis virus (VEE) vectors. For example, MV-CEA was administered intraperitoneally in a phase I/II trial in 21 patients with platinum- and paclitaxel-refractory ovarian cancer in the peritoneal cavity [76]. No dose-limiting toxicity was discovered with only mild treatment-related adverse events. Stable disease (SD) was achieved in 9 out of 9 patients receiving a high dose of MV-CEA compared to 5 out of 12 immunized with a lower dose [76]. The median overall survival (OS) was 12.15 months, which was twice as long as the expected OS. Moreover, for patients receiving the high dose of MV-CEA, the median OS was 38.4 months. In another phase I clinical study, the treatment of patients with recurrent glioblastoma multiforme with MV-CEA is currently in progress [77,78] in a phase I trial in patients with stages III and IV colorectal cancer. VEE-CEA particles were administered four times every three weeks [79]. The outcomes of the study were antigen-specific immune responses and extended overall survival in both stage III and IV patients. In another phase I trial, VEE-CEA particles were intramuscularly administered to patients with metastatic pancreatic cancer [80]. Immunization with VEE-CE A induced clinically relevant T-cell and antibody responses, cellular toxicity in tumor cells, and prolonged OS.

In a phase I study in five patients with cutaneous T-cell lymphomas (CTCLs) intratumoral administration of the oncolytic MV-Edmonston-Zagreb strain (MV-EZ) showed good safety and tolerance [81]. Complete regression of one CTCL tumor was registered in one patient after the first treatment cycle. Moreover, partial regression was detected in four out of five treated tumors. In another study, the MV-NIS vector was intraperitoneally administered to patients with recurrent ovarian cancer in a phase I trial [82]. There were only mild adverse events and no dose-limiting toxicity in the 16 platinum-resistant and heavily pretreated patients enrolled in the study. SD was obtained in 13 out of 16 patients, and the median OS was 26.6 months, significantly longer than the expected median survival of 6–12 months. Additionally, phase I studies using MV-NIS on pleural mesothelioma [83], malignant peripheral nerve sheath tumors [84], and advanced recurrent or metastatic head and neck and breast cancers [85] are in progress. In another phase I trial in patients with multiple myeloma, complete remission of one patient was achieved after MV-NIS administration [86]. The patient showed strong baseline T-cell responses both to MV proteins and to 8 out of 10 tested TAAs. 

A phase I trial in patients with castration-resistant metastatic prostate cancer (CRPC) has been conducted with VEE-PSMA particles [87]. Although the treatment showed good safety and tolerance, the PSMA-specific immune responses were disappointingly weak. In another phase I trial, stage IV HER2-overexpressing breast cancer patients were vaccinated with VEE-HER2 particles [88]. The VEE-HER2 particle administration was safe and well tolerated. SD was achieved in two patients and partial response (PR) in one patient. Although positive results concerning safety and efficacy have been achieved in clinical trials, no FDA-approved drugs based on oncolytic self-replicating RNA viruses are approved today. However, Ervebo, the VSV-based vaccine against Ebola virus disease, has been approved both by the FDA and the EMA [89]. 

## 5. Conclusions

In this review, examples of preclinical studies and clinical trials applying oncolytic self-replicating RNA viruses for the treatment and prevention of various cancers have been presented. In many cases, inhibition of tumor growth, eradication of tumors, and even cure have been observed. Similar to other types of oncolytic viruses, the specific replication in tumor cells leading to their killing without causing much damage to normal cells has been described. The extensive self-replication of viral RNA in the cytoplasm of infected tumor cells potentially improves the oncolytic activity and reduces the doses required for therapeutic efficacy. Moreover, self-replicating RNA viruses are known to induce apoptosis in infected cells [90], providing additional therapeutic potential contributing to the killing of tumor cells. One feature of interest of self-replicating RNA viruses comprises their flexibility to be delivered as replication-proficient oncolytic viruses, replication-deficient recombinant particles, RNA replicons, or plasmid-based DNA replicons. As described above, DNA-based delivery of VSV vectors generated significant tumor regression and prolonged survival in ovarian mouse tumor models [71,72]. However, application of DNA-based vectors requires delivery to the nucleus, which is less efficient than the cytoplasmic delivery of RNA. Nonetheless, the drawback of using RNA-based delivery is the degradation sensitivity of ssRNA molecules, which has triggered the engineering RNA molecules less prone to degradation and nanoparticle formulations for improved delivery and protection against cellular RNases. Another issue of concern when viral vectors are readministered relates to the negative effect on transgene expression and antitumor activity caused by antiviral immune responses, which has been frequently seen after repeated administration of adeno-associated viruses (AAVs) [91]. However, several studies have demonstrated that self-replicating viruses, especially alphaviruses, show relatively weak antiviral immunogenicity, demonstrating only a minor effect on the therapeutic antitumor activity [19]. For example, a booster immunization of mice with SFV particles expressing the human papilloma virus (HPV) E6 and E7 genes did not show reduced antigen production by SFV-specific antibodies, inhibition of CTL responses, or impediment of transgene-specific responses in a mouse model [92]. Moreover, VEE-CEA particles, which efficiently infected dendritic cells (DCs), could be readministered to patients with metastatic tumors, generating clinically relevant CEA-specific T-cell responses [80].

Although relevant therapeutic efficacy of self-replicating RNA viruses has been achieved in various animal models for different cancers, and good safety and tolerance have been demonstrated, the therapeutic success rate has been more modest in clinical trials. Obviously, the number of clinical studies is much lower than proof-of-concept studies in preclinical settings. However, the transition to human use has been problematic for the following reasons. Similar to findings related to other viral and nonviral delivery systems, the transition from studies in relatively small rodents to much larger human beings has not been straightforward. The other point of importance relates to the application of xenograft models of induced tumors in rodents, which do not accurately resemble spontaneous cancers occurring in humans. For these reasons, canine models can provide an important and more relevant approach due to the larger size of the subjects studied and also the presence of spontaneously occurring natural cancers, as described for the successful tumor eradication by oncolytic ZIKV in dogs [45]. One area of utilization of oncolytic viruses relates to combination therapy with drugs or other viruses. For example, combining the oncolytic M1 alphavirus with doxorubicin enhanced tumor growth inhibition 100-fold in mice [39]. Likewise, ZIKV and anti-PD1 coadministration provided a synergistic effect on survival in a glioblastoma mouse model [43]. A superior tumor mass reduction was seen in pancreatic tumor cells after MV and gemcitabine combination therapy [58]. Administration of the oncolytic Maraba virus induced antigen-specific immune responses in mice with B16-F10 melanoma metastases after prime immunization with an adenovirus vector [74]. Furthermore, coadministration of oncolytic MV and MuV vectors provided superior antitumor activity in a prostate tumor mouse model [62].

In conclusion, although progress has been seen for therapeutic applications of oncolytic self-replicating RNA viruses, further development is necessary to realize the full potential of these attractive gene therapy vectors. 

## Figures and Tables

**Figure 1 ijms-23-15622-f001:**
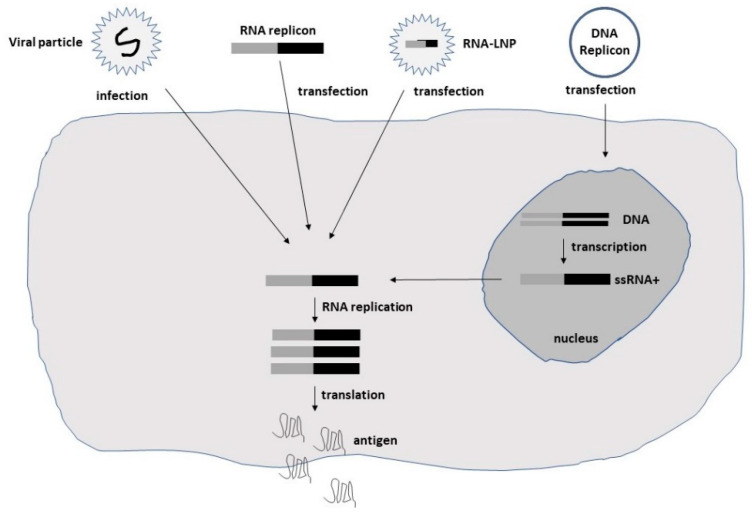
Schematic illustration of the delivery of self-replicating RNA viruses. Viral particles, naked RNA replicons, lipid nanoparticle (LNP)-encapsulated RNA, or DNA replicons can be used.

**Table 1 ijms-23-15622-t001:** Examples of preclinical studies using oncolytic self-replicating RNA viruses.

Cancer	Oncolytic Virus	Gene(s)	Findings	Ref.
**Alphaviruses**				
GBM	SFV VA7	EGFP, Rluc	Tumor eradication, long-term survival in mice	[31]
Lung A459	SFV-VA7	EGFP	Prolonged survival in mice	[26]
Prostate LNCaP	SFV-VA7	EGFP	Tumor cell killing, tumor eradication in mice	[32]
GBM	SFV-AM6-124T	miR124	Targeting GL261 gliomas, enhanced by anti-PD1	[33]
GBM	SFV4miRT	miR124,125,134	Prolonged survival in mice	[34]
Cervical	SIN AR339	SIN AR339	Tumor cell killing, tumor regression in mice	[25]
Ovarian	SIN AR339	SIN AR339	Tumor cell killing, tumor regression in mice	[25]
Liver	M1	GFP	Targeting of liver tumors in mice	[35]
Glioma	M1	M1	Killing of malignant glioma cells in mice, rats	[36]
Bladder MIBC	M1	GFP	Tumor growth inhibition in mice	[37]
Bladder	M1	M1	Oncolytic activity in mouse bladder tumor model	[38]
Breast TNBC	M1	M1 + Dox	Reduced tumor growth in mice	[39]
Pancreatic	M1	M1 + IRE	Superior tumor inhibition, prolonged survival	[40]
Liver	M1	M-LPO	Inhibition of Hep3B cancer cell growth in vitro	[41]
Colorectal	M1	M-LPO	Inhibition of LoVo cancer cell growth in vitro	[41]
**Flaviviruses**				
GBM	ZIKV	m-ZIKV	Prolonged survival in mice	[28]
MB, ependymoma	ZIKV	ZIKV	Infection and killing of GSCs	[42]
GBM	ZIKV	ZIKV + anti-PD1	Synergistic effect on survival in mice	[43]
Embryonal CNS	ZIKV	ZIKVBR	Eradication of brain tumors, no effect on normal cells	[44]
Spontaneous CNS	ZIKV	ZIKVBR	Tumor eradication, prolonged survival in dogs	[45]
Prostate	ZIKV	ZVp	Metabolomics to identify PC-3 cancer cell markers	[46]
**Measles viruses**				
Medulloblastoma	MV	GFP	Complete tumor regression in mice	[47]
Medulloblastoma	MV	GFP	Significantly prolonged survival in mice	[48]
Glioma	MV	CEA, NIS	Cytopathic effects in GSC cell lines	[49]
Breast	MV	SLAMblind	Anti-tumor activity in mice	[50]
Breast	MV	MV	Infection, killing of MCF-7 and CAL-51 cancer cells	[51]
Breast	MV	MV-Edm	Re-sensitization of Dox and ironicizing radiation	[52]
Lung	MV	MV Hu-191	Suppression of tumor growth in mice	[53]
Lung. colorectal	MV	MV-Schwarz	Repression of tumor growth in mice	[54]
Lung	MV	CEA	Tumor growth inhibition in mice	[55]
Melanoma	MV	MV L-16	Killing of tumor cells, tumor inhibition in mice	[56]
Pancreatic	MV	SLAMBlind	Inhibition of tumor growth in mice	[57]
Pancreatic	MV	MV-SCD + Gem	Reduced tumor mass in pancreatic cell lines	[58]
Pancreatic	MV	MV-miR-148	Delayed tumor growth, prolonged survival in mice	[59]
Prostate	MV	CEA	Delayed tumor growth, prolonged survival in mice	[60]
Prostate	MV	sc-Fv-PSMA	Killing of prostate cancer cells	[61]
Prostate	MV	MV + MuV	Superior anti-tumor activity, survival in mice	[62]
**Rhabdoviruses**				
Glioma, breast	VSV	VSVrp30a	Targeting and eradication of tumors in mice	[63]
Olfactory bulb	VSV	VSVrp30a	Tumor targeting, no damage to normal cells in mice	[63]
Glioblastoma	VSV	VSV-p1-GFP	Killing of tumor cells, not normal cells	[64]
Breast 4T1	VSV	VSV(M51R)-LacZ	Lesions in breast cancer cells in mice	[65]
Colon CT-26	VSV	VSV(M51R)	Prolonged survival in mice	[66]
Lung LLC-1	VSV	VSV-LCMV GP	Tumor-to-tumor spread, killing of tumor cells	[67]
Melanoma	VSV	VSV-LCMV GP	Tumor regression, prolonged survival in mice	[68]
Ovarian	VSV	VSV-LCMV GP	Superior tumor regression with ruxolitinib	[69]
Melanoma	VSV	VSV-XN2-ΔG	Tumor regression in mice	[70]
Ovarian	VSV	VSVMP-p DNA	Tumor weight decrease, prolonged survival in mice	[71]
Ovarian	VSV	VSVMP-p DNA	87–98% tumor regression, prolonged survival	[72]
Prostate	VSV	VSV(M51R)	Superior oncolysis after curcumin treatment	[73]
Melanoma	Maraba MG1	hDCT + Ad-hDCT	Immune response after prime Ad-hCDT	[74]
Sarcoma	Maraba MG1	MG1	Protection against tumor challenges, cure in mice	[30]
Breast	MV, RABV	rMVEGFP-LDMV	Blue light induced tumor regression	[75]

Ad-hCT, adenovirus hDCT; anti-PD1, anti-programmed death 1; CEA, carcinoembryonic antigen; CNS, central nervous system; Dox, doxorubicin; EGFP, enhanced green fluorescent protein; GBM, glioblastoma multiforme; Gem, gemcitabine; GSCs, glioblastoma stem cells; hDCT, human dopachrome tautomerase; IRE, irreversible electroporation; LCMV GP, lymphocytic choriomeningitis virus glycoprotein; M1, M1 alphavirus; miRNA, microRNA; MB, medulloblastoma; M-LPO, liposome encapsulated M1; MuV, mumps virus; MV, measles virus; MV L-16, MV Leningrad-16 strain: m-ZIKV, mouse adapted ZIKV; NIS, sodium iodide symporter; PSMA, prostate-specific membrane antigen; RABV, rabies virus; rMV^EGFP^-L_DMV_, MV with EGFP and controllable Magnet; Rluc, Renilla luciferase; sc-Fv, single-chain antibody; SFV, Semliki Forest virus; SIN, Sindbis virus; SLAMBlind, disenabled signaling lymphocyte activation molecule; TNBC, triple-negative breast cancer; VSV, vesicular stomatitis virus; VSV(M51R), VSV with mutation in matrix protein; VSVMP-p, liposome encapsulated VSV DNA vector with matrix protein ZIKV, Zika virus; ZIKV^BR^, Brazilian ZIKV strain; ZVp, inactivated ZIKV prototype.

**Table 2 ijms-23-15622-t002:** Examples of clinical studies using oncolytic self-replicating RNA viruses.

Cancer	Oncolytic Virus	Phase	Findings	Ref
Ovarian	MV-CEA	I/II	No toxicity, SD in patients, 2-fold extended OS	[76]
GBM	MV-CEA	I	Study in progress	[77,78]
Colorectal	VEE-CEA	I	Antigen-specific responses, extended survival	[79]
Pancreatic	VEE-CEA	I	T cell responses, tumor toxicity, extended OS	[80]
CTCL	MV-EZ	I	Good safety, complete tumor regression	[81]
Ovarian	MV-NIS	I	SD in patients, significantly extended OS	[82]
Mesothelioma	MV-NIS	I	Study in progress	[83]
MPNST	MV-NIS	I	Study in progress	[84]
Head & Neck	MV-NIS	I	Study in progress	[85]
Myeloma	MV.NIS	I	Complete remission in one patient	[86]
Prostate	VEE-PSMA	I	Safe, but disappointingly weak immune response	[87]
Breast	VEE-HER2	I	SD in 1 patient, PR in 2 patients	[88]

CEA, carcinoembryonic antigen; CTCL, cutaneous T-cell lymphoma; GBM, glioblastoma multiforme; HER2, human epidermal growth factor receptor 2; MPNSST, malignant peripheral nerve sheath tumor; MV, measles virus; MV-EZ, MV Edmonston-Zagreb strain; NIS, sodium iodide symporter; OS, overall survival; PR, partial response; PSMA, prostate-specific membrane antigen; SD, stable disease; VEE, Venezuelan equine encephalitis virus.

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
