# Peer review of "Therapeutic Applications for Oncolytic Self-Replicating RNA Viruses"

_ijms, 2022, doi:10.3390/ijms232415622_

Round 1

Reviewer 1 Report

This review is addressed to  self-replicating RNA viruses  that have become an attractive tool for anti-cancer therapy. The review is undoubtedly relevant and interesting for a wide range of specialists.

 My comments and suggestion. 

1.      The review is hard to read, it is overloaded with facts and depleted in their understanding. It remains unclear: what the mechanism of selectivity of the antitumor virus  cytotoxicity;  how the effects of some viruses differs from the effects of other viruses, what  the relationships between the direct cytotoxic antitumor  effect of the virus and its indirect immune-mediated action …

2 2.  It should be pointed out that the antiviral immune response sharply reduces the antitumor efficacy of repeated virus applications.

3 3.  An abstract would be appropriate to reflect more widely available clinical data.

4 4.  In the     conclusions” section  it would be useful to outline the possible place of viral therapy in the complex (multiple-purpose) cancer  treatment.

Author Response

This review is addressed to  self-replicating RNA viruses  that have become an attractive tool for anti-cancer therapy. The review is undoubtedly relevant and interesting for a wide range of specialists.

 My comments and suggestion. 

  1. The review is hard to read, it is overloaded with facts and depleted in their understanding. It remains unclear: what the mechanism of selectivity of the antitumor virus  cytotoxicity;  how the effects of some viruses differs from the effects of other viruses, what  the relationships between the direct cytotoxic antitumor  effect of the virus and its indirect immune-mediated action …

Response: I understand that there are plenty of facts but is it not the purpose of reviews to provide plenty of information! To “lighten” the review a bit in addition to Table 1, Table 2 has been added as well as Fig. 1 illustrating the delivery methods developed for oncolytic self-replicating RNA viruses.   

  1. It should be pointed out that the antiviral immune response sharply reduces the antitumor efficacy of repeated virus applications.

Response: This point has now been addressed in the Conclusions section.

  1. An abstract would be appropriate to reflect more widely available clinical data.

Response: I respectfully disagree with this comment as in the current Abstract almost one third (61/199 words) deals with clinical trials.

  1. In the “conclusions” section  it would be useful to outline the possible place of viral therapy in the complex (multiple-purpose) cancer treatment.

Response: The point suggested by the reviewer has now been discussed in the Conclusions section.

Reviewer 2 Report

* The author reviewed an interesting topic regarding Oncolytic Self-Replicating RNA Viruses but some moderate revision by a native English speaker or a specialized editing company is required. I have some issues:

* Please, make a table for the  clinical trails.

* Please, make some figures explaining the different mechanisms of drug delivary and interacting of Oncolytic Self-Replicating RNA Viruses with the cancer cells.

* Please, make a paragraph with table showing and discussing the various international patents regarding Oncolytic Self-Replicating RNA Viruses.

* Please, mention any FDA approved drugs regarding Oncolytic Self-Replicating RNA Viruses and if there is no any approved drugs, please, mention it.

Author Response

Comments and Suggestions for Authors

* The author reviewed an interesting topic regarding Oncolytic Self-Replicating RNA Viruses but some moderate revision by a native English speaker or a specialized editing company is required. I have some issues:

Response: The manuscript has now been checked and revised by an eminent English-speaking scientist with the experience of more than 300 peer-reviewed publications.

* Please, make a table for the  clinical trails.

Response: Thank you for the suggestion! Table 2 with examples of clinical trials conducted with oncolytic self-replicating RNA viruses has been added.

* Please, make some figures explaining the different mechanisms of drug delivary and interacting of Oncolytic Self-Replicating RNA Viruses with the cancer cells.

Response: Thank you for the suggestion! Figure 1 describing the delivery of oncolytic self-replicating RNA viruses has been added.

* Please, make a paragraph with table showing and discussing the various international patents regarding Oncolytic Self-Replicating RNA Viruses.

Response: I respectfully disagree with this suggestion. Although a table on patents could be useful under some circumstances, I feel it not under the scope of this review. 

* Please, mention any FDA approved drugs regarding Oncolytic Self-Replicating RNA Viruses and if there is no any approved drugs, please, mention it.

Response: Text has been added to the end of section 4 stating that no FDA approved drugs are yet available but mentioning that the VSV-based vaccine Ervebo against Ebola virus disease ahs been approved.

Round 2

Reviewer 1 Report

The author has improved the article. I have no more comments or suggestions.

Author Response

OK, thanks

Reviewer 2 Report

* The authors made good efforts in the improving the manuscript quality. I do not understand the reason for non suitability of patents writing. 

Author Response

 I still stand by my point that a paragraph and table on patents would make the review unnecessarily long (Reviewer 2 already complained that "the review is hard to read and overloaded with facts". Moreover, it would not add much to the therapeutic applications and therefore I do not consider it to be the focus of the review. Patent business is a major field of its own. It would be much better to separately write a review on "Patents on Self-replicating RNA Viruses and Their Applications", which I would be happy to consider if IJMS would be interested.